# Association between serum β2-microglobulin and mortality in Japanese peritoneal dialysis patients: A cohort study

Yukio Maruyama[1,2]*, Masaaki Nakayama[3], Masanori Abe[2,4], Takashi Yokoo[1], Jun Minakuchi[5], Kosaku Nitta[2,6]

1 Division of Nephrology and Hypertension, Department of Internal Medicine, The Jikei University School of Medicine, Tokyo, Japan, 2 Committee of Renal Data Registry, Japanese Society for Dialysis Therapy, Tokyo, Japan, 3 Department of Nephrology, St. Luke's International Hospital, Tokyo, Japan, 4 Division of Nephrology, Hypertension and Endocrinology, Department of Internal Medicine, Nihon University School of Medicine, Tokyo, Japan, 5 Department of Kidney Disease, Kawashima Hospital, Tokushima, Japan, 6 Fourth Department of Internal Medicine, Tokyo Women's Medical University, Tokyo, Japan

* maruyama@td5.so-net.ne.jp

**Data Availability Statement:** The data underlying the results presented in the study are available from the Japanese Society for Dialysis Therapy (JSDT). Contact information of the Committee of

## Abstract

### Background

Higher serum β2-microglobulin (B2M) concentrations are associated with higher mortality in the general population, non-dialyzed chronic kidney disease patients and patients receiving hemodialysis (HD). However, this relationship among patients on peritoneal dialysis (PD) has not been validated.

### Methods

We collected baseline data for 3,011 prevalent PD patients from a nationwide dialysis registry in Japan at the end of 2010. Clinical outcomes for 9 years were then evaluated using the registry at the end of 2011 to 2019. All-cause and cardiovascular (CV) mortality was assessed using Cox regression analysis and competing-risks regression analysis, respectively. We used multiple imputation to deal with missing covariate data.

### Results

During a median follow-up of 87 months, 2,054 patients transferred to combined therapy with PD and HD or HD directly. A total of 3,011 patients, 1,235 (41.0%) died, including 437 patients (14.5%) from CV causes. Among them, 612 patients died after transfer to other dialysis modalities. Univariate analyses revealed no significant association between serum B2M and mortality, whereas higher serum B2M was independently associated with both all-cause and CV mortalities in adjusted models. However, the significant association between serum B2M and CV mortality disappeared in analysis treating serum B2M as a categorical variable. The effect of serum B2M on all-cause mortality was significantly higher among patients with higher urinary volume and a significant interaction was evident.

Renal Data Registry, Japanese Society for Dialysis Therapy is toukei@jsdt.or.jp.

**Funding:** This research received no specific grant from any funding agency in the public, commercial, or not-for-profit sectors. The funders had no role in study design, data collection and analysis, decision to publish, or preparation of the manuscript.

**Competing interests:** Y.M. received honoraria from Baxter International, Inc. and Terumo Corporation without any other relevant declarations relating to employment, consultancy, patents, products in development, marketed products, etc. This research received no specific grant from any funding agency in the public, commercial, or not-for-profit sectors. This does not alter our adherence to PLOS ONE policies on sharing data and materials. No other authors have any conflicts of interest to declare.

## Conclusions

Using a large-scale registry, we found that serum B2M contributes tenuously but significantly to worse outcome and residual kidney function significantly affects this relationship. On the contrary, serum B2M *per se* had no predictive value for patient outcome in prevalent PD patients.

## Introduction

In the management of peritoneal dialysis (PD), solute and water removal are monitored to decide the appropriate modality of dialysis. Assessment of solute removal is largely dependent on the measurement of small solute clearance, including Kt/V and creatinine clearance [1]. Although the total Kt/V urea was recommended to be maintained at 1.7 according to a previous clinical guideline [1], a more recent guideline did not establish target small solute clearance, because only a low level of evidence suggests that increasing urea clearance has any impact on quality of life, technique survival or mortality [2].

As a middle molecular weight uremic toxin, β2-microglobulin (B2M) (11,800 Da) is produced by all cells expressing major histocompatibility class I. This protein is the major protein component of dialysis-related amyloidosis [3]. Since B2M is removed exclusively by the kidneys, concentration in the body increase in parallel with declines in glomerular filtration rate (GFR) in chronic kidney disease (CKD) and reach the highest levels among dialyzed patients [4–8]. Serum B2M level is also associated with several comorbid conditions, such as malignancy and inflammation. Higher serum B2M is well-known to be associated with higher mortality and rapid declines in kidney function among the general population [9, 10], non-dialyzed CKD patients [11] and patients on hemodialysis (HD) [12]. Indeed, the European Best Practice Guidelines have recommended the use of B2M as a marker for middle molecular weight uremic toxin and to maximize its removal in HD patients [13].

Serum B2M is also used as indicator of dialysis efficiency in PD patients. Notably, the efficiency of B2M removal is much lower in PD than in HD or on-line hemodiafiltration [14]. On the other hand, PD patients tend to maintain renal kidney function (RKF), as an important contributor to patient survival, more than HD patients [15]. B2M can accurately estimate RKF in dialyzed patients [7, 8, 16, 17]. In addition, we reported that serum B2M was higher in patients with a history of encapsulating peritoneal sclerosis, one of the most serious complications in PD therapy, as compared to those without [18, 19]. Serum B2M levels may thus have different impacts on patient outcome in PD patients from those in HD patients. Although a few reports have investigated the effects of serum B2M on mortality in PD patients, results have not been consistent, partially due to limited numbers of enrolled patients and shorter observational periods [20, 21].

The aim of this study was to investigate the impact of serum B2M levels on patient survival in PD patients, using data from a large-scale registry of dialysis patients in Japan.

## Materials and methods

### Study design and patient selection

The Japanese Society for Dialysis Therapy (JSDT) has conducted annual surveys of dialysis facilities throughout Japan. These surveys address epidemiological backgrounds, treatment conditions, and outcomes of treatment with dialysis. At the end of 2010, a total of 9,773

patients were undergoing PD in Japan [22]. Among these, we extracted 8,709 patients who were 18 years and older and had been undergoing PD for more than 3 months. The Medical Ethics Committee of the JSDT approved the study protocol (approval number: Dan-4, the date of approval: July 2, 2020). From an ethical perspective, all data were fully anonymized before being accessed. The requirement to obtain informed consent from the patients was waived because of the retrospective nature of the study. Instead, all individual participants were provided the opportunity to opt out of this study.

## Measurements

Biochemical parameters including B2M, blood urea nitrogen (BUN), creatinine (Cr), serum albumin (Alb), C-reactive protein (CRP) and hemoglobin (Hb) were measured using standard laboratory techniques at each facility. We divided baseline B2M into four categories according to quartiles: <18.5 mg/L, 18.5 to <24.7 mg/L, 24.7 to <33.3 mg/L, and ≥33.3 mg/L. The dialysate-to-plasma ratio of creatinine (D/P Cr), obtained from a peritoneal equilibration test (PET), and total, renal, and PD Kt/V at initiation of PD were calculated at each facility [23].

## Outcomes

Primary and secondary outcome was 9-year all-cause death and death from cardiovascular (CV) disease, respectively. This information was extracted from the data at the end of 2011 to 2019. CV death was defined as death caused by heart failure, acute myocardial infarction, arrhythmia, valvular disease, subarachnoid hemorrhage, cerebral hemorrhage, cerebral infarction, or sudden death.

## Statistical analysis

Data are presented as mean ± standard deviation or median and interquartile range. Patients with and without serum B2M data were compared using Student's t-test, the Wilcoxon rank-sum test, or the chi-square test, as appropriate. Subsequently, characteristics of the population were categorized by quartiles of serum B2M, and were compared by one-way analysis of variance or the non-parametric Kruskal-Wallis test for continuous variables, and the chi-square test for nominal variables. Multiple regression analysis was used to evaluate independent factors affecting serum B2M. Kaplan-Meier survival analysis was used to compare all-cause and CV mortality between patient groups divided according to serum B2M. Hazard ratios (HRs) and 95% confidence intervals (CIs) for all-cause death was assessed using Cox regression analysis. Standard sub-hazard ratios (SHRs) and 95%CIs for CV death were assessed using competing-risks regression analysis and considering non-CV death as the competing event. Competing-risks regression analysis was based on Fine and Gray's proportional sub-hazard model [24] and modified for a STATA-specific presentation [25]. In these multivariate analyses, covariates were sex, age, dialysis duration, original disease, body mass index (BMI), laboratory data including BUN, Cr, Alb, CRP and Hb, comorbid disease and factors associated with PD managements including D/P Cr ratio, use of icodextrin, urine volume (UV) and history of PD peritonitis. Multiple imputation for missing values was performed in these multivariate analyses. The missing values of all covariates were imputed, assuming data were missing at random, with 20 imputations [26]. B2M, dialysis duration, CRP, and UV were markedly skewed, and log-transformed to normalize the distributions before analysis. We additionally explored the continuous, potentially nonlinear relationship between serum B2M and all-cause mortality by using restricted cubic spline models with five knots at the 5th, 27.5th, 50th, 72.5th, and 95th percentiles. Subgroup analyses were performed for all baseline covariates including sex, age, dialysis duration, original disease, BMI, laboratory data, comorbid disease

and factors associated with PD management at the beginning of follow-up. All covariates were divided into categorical groups and continuous variables were divided by medians except for UV. UV was divided into two groups about a cut-off value of 100 mL/day. Data were statistically analyzed using STATA version 16.0 (STATA Corporation, College Station, TX, USA). Values of P < 0.05 were considered significant.

## Results

Fig 1 summarizes the process of data extraction. The original data set included a total of 8,709 patients aged 18 years or older and on PD for more than 3 months as of the end of 2010. We excluded 1,826 patients on combined therapy with PD and HD. For the remaining 6,883 patients, we first compared baseline characteristics between patients with and without serum B2M data (S1 Table). Data on serum B2M were available for 3,011 patients. Patients with B2M data displayed a higher prevalence of chronic glomerulonephritis (CGN) and nephrosclerosis as underlying diseases, fewer comorbidities of cerebral infarction, higher BUN, higher Cr, higher Alb, higher Hb and lower PD Kt/V as compared to those without B2M data. Of note, no effect on crude all-cause mortality was seen for the presence or absence of serum B2M data (HR 0.99; 95%CI, 0.92 to 1.06).

Table 1 shows baseline characteristics of 3,011 patients (age, 63 ± 13 years; male, 61.4%; median dialysis duration, 31 months) with data for serum B2M. Underlying pathologies comprised CGN in 1,301 patients (43.2%), diabetic nephropathy in 856 (28.4%), nephrosclerosis in

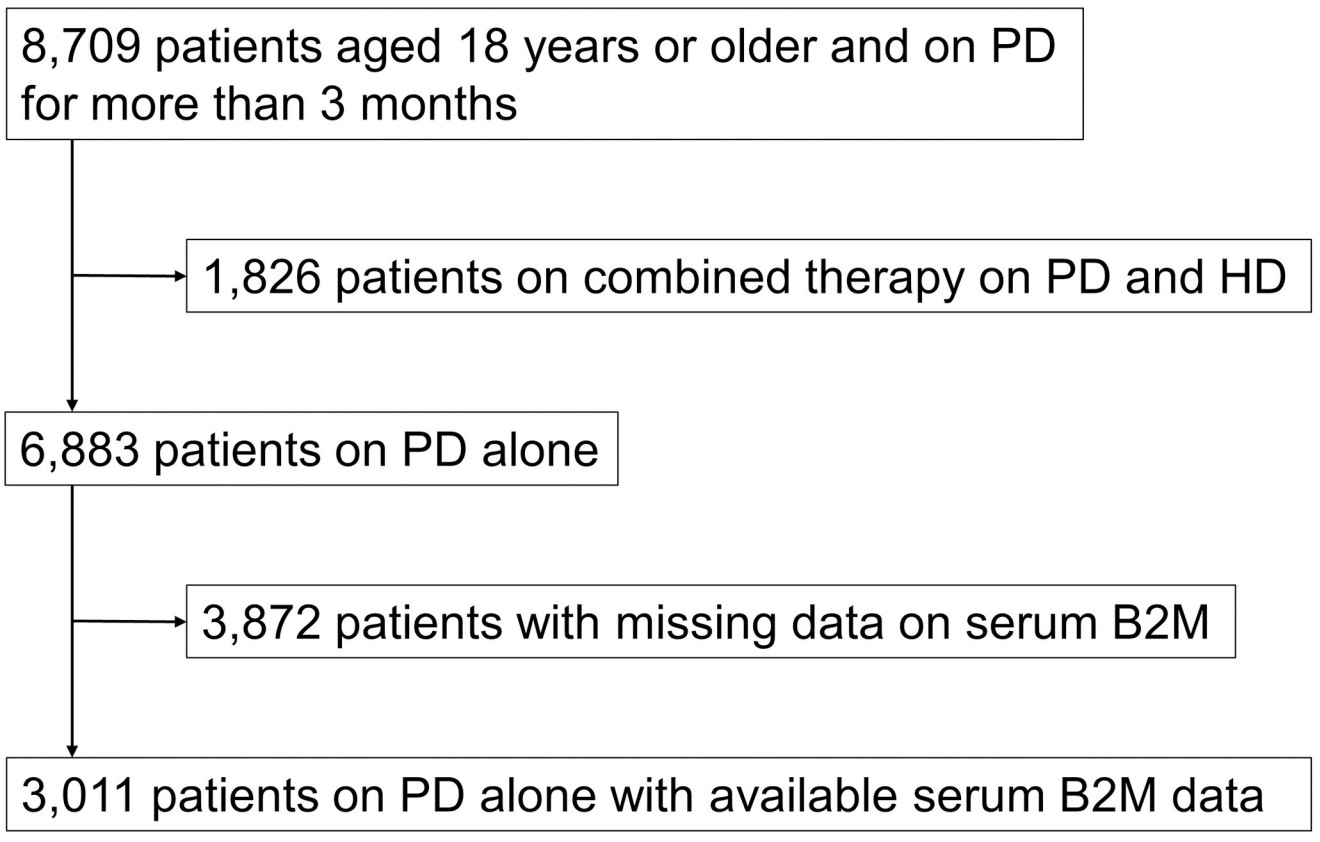

**Fig 1. Patient selection flowchart.** Abbreviations: PD, peritoneal dialysis; HD, hemodialysis; B2M, β2 microglobulin.

**Table 1. Baseline characteristics of 3011 PD patients by quartiles of serum B2M.**

| Variable | No. of missing value (%) | Whole group | B2M quartiles (mg/L) | | | | P for trend |
|---|---|---|---|---|---|---|---|
| | | | < 18.5 | 18.5 to < 24.7 | 24.7 to < 33.3 | ≥ 33.3 | |
| Number (%) | | 3011 | 1188 (25.1%) | 1126 (23.7%) | 1176 (24.8%) | 1252 (26.4%) | |
| Age (years) | 0 (0%) | 63±13 | 64±14 | 64±13 | 62±13 | 60±13 | <0.01 |
| Male (%) | 0 (0%) | 1848 (61.4%) | 494 (65.5%) | 458 (60.6%) | 450 (60.0%) | 446 (59.4% | 0.06 |
| Dialysis duration (months) | 0 (0%) | 31 (15–54) | 17 (9–31) | 23 (12–41) | 35 (20–55) | 55 (33–90) | <0.01 |
| Height (cm) | 338 (11.2%) | 160±9 | 160±9 | 160±9 | 160±10 | 161±9 | 0.69 |
| BW (kg) | 467 (15.5%) | 60.1±12.4 | 60.5±12.6 | 59.7±11.8 | 60.4±12.8 | 59.7±12.5 | 0.43 |
| BMI (kg/m$^2$) | 649 (21.6%) | 23.2±3.7 | 23.3±3.6 | 23.1±3.6 | 23.3±3.8 | 23.0±3.9 | 0.39 |
| Underlying disease | 0 (0%) | | | | | | <0.01 |
| CGN (%) | | 1301 (43.2%) | 278 (36.9%) | 331 (43.8%) | 313 (41.7%) | 379 (50.5%) | |
| Diabetic nephropathy (%) | | 856 (28.4%) | 229 (30.4%) | 209 (27.6%) | 228 (30.4%) | 190 (25.3%) | |
| Nephrosclerosis (%) | | 362 (12.0%) | 119 (15.8%) | 98 (13.0%) | 83 (11.1%) | 62 (8.3%) | |
| PKD (%) | | 76 (2.5%) | 19 (2.5%) | 19 (2.5%) | 18 (2.4%) | 20 (2.7%) | |
| Others or unknown (%) | | 416 (13.8%) | 109 (15.5%) | 99 (13.1%) | 108 (14.4%) | 100 (13.3%) | |
| Comorbidity | | | | | | | |
| AMI | 260 (8.6%) | 182 (6.6%) | 49 (7.1%) | 54 (7.7%) | 33 (4.8%) | 46 (6.8%) | 0.15 |
| Cerebral hemorrhage | 261 (8.7%) | 88 (3.2%) | 14 (2.1%) | 26 (3.7%) | 22 (3.2%) | 26 (3.8%) | 0.22 |
| Cerebral infarction | 259 (8.6%) | 290 (10.5%) | 64 (9.3%) | 77 (11.0%) | 70 (10.2%) | 79 (11.6%) | 0.53 |
| Quadruple amputation | 245 (8.1%) | 25 (0.9%) | 6 (0.9%) | 4 (0.6%) | 7 (1.0%) | 8 (1.2%) | 0.67 |
| Laboratory data | | | | | | | |
| B2M (mg/L) | 0 (0%) | 24.6 (18.4–33.2) | 15.5 (12.9–17.0) | 21.2 (19.9–22.9) | 28.6 (26.4–30.8) | 39.2 (36.0–44.1) | <0.01 |
| BUN (mg/dL) | 21 (0.7%) | 54±15 | 53±16 | 57±14 | 57±15 | 54±14 | <0.01 |
| Cr (mg/dL) | 15 (0.5%) | 9.6±3.3 | 7.0±2.4 | 9.1±2.5 | 10.6±2.9 | 11.9±3.1 | <0.01 |
| Alb (g/L) | 40 (1.3%) | 3.4±0.5 | 3.5±0.5 | 3.4±0.5 | 3.3±0.5 | 3.4±0.6 | <0.01 |
| CRP (mg/dL) | 357 (11.9%) | 0.1 (0–0.5) | 0.1 (0–0.3) | 0.1 (0–0.4) | 0.1 (0–0.5) | 0.2 (0–0.6) | <0.01 |
| Hb (g/dL) | 33 (1.1%) | 10.4±1.5 | 10.9±1.5 | 10.5±1.4 | 10.2±1.3 | 10.2±1.5 | <0.01 |
| PD related parameters | | | | | | | |
| D/P Cr | 1471 (48.9%) | 0.66±0.13 | 0.64±0.14 | 0.66±0.13 | 0.69±0.13 | 0.65±0.13 | <0.01 |
| Use of icodextrin | 569 (18.9%) | 533 (21.8%) | 61 (9.7%) | 91 (14.4%) | 156 (26.4%) | 225 (38.3%) | <0.01 |
| UV (mL/day) | 1076 (35.7%) | 650 (200–1030) | 1010 (790–1430) | 810 (550–1200) | 500 (150–800) | 50 (0–300) | <0.01 |
| Renal Kt/V | 1871 (62.1%) | 0.5 (0.1–0.8) | 0.9 (0.6–1.3) | 0.6 (0.4–0.9) | 0.3 (0.1–0.6) | 0 (0–0.2) | <0.01 |
| PD Kt/V | 1658 (55.1%) | 1.3 (1.0–1.7) | 1.1 (0.7–1.6) | 1.2 (0.9–1.5) | 1.4 (1.1–1.7) | 1.6 (1.4–1.9) | 0.01 |
| History of PD peritonitis | 779 (25.9%) | 387 (17.3%) | 74 (13.3%) | 106 (18.2%) | 101 (18.5%) | 106 (19.4%) | 0.03 |

Date is shown as means ± SD or medians and interquartile ranges (IQR).

Abbreviations: PD, peritoneal dialysis; B2M, β2 microglobulin; BW, body weight; BMI, body mass index; CGN, chronic glomerulonephritis; PKD, polycystic kidney disease; AMI, acute myocardial infarction; BUN, blood urea nitrogen; Cr, creatinine; Alb, albumin; CRP, C-reactive protein; Hb, hemoglobin; D/P Cr, dialysate-to-plasma ratio of creatinine; UV, urinary volume.

362 (12.0%), polycystic kidney disease in 76 (2.5%), and other or unknown in 416 (13.8%). Patients with higher B2M were younger, had longer duration of dialysis, higher Cr, higher CRP, lower Hb, lower UV, lower renal Kt/V and higher PD Kt/V. Regarding underlying disease, CGN was more prevalent among the higher B2M group, whereas diabetic nephropathy and nephrosclerosis were more prevalent among the lower B2M group. Icodextrin users and patients with a history of PD-associated peritonitis were more prevalent among the higher B2M group.

**Table 2. Contributing factors for serum B2M.**

| Variable | Unadjusted | | | | Adjusted | | | |
|---|---|---|---|---|---|---|---|---|
| | Regression Coefficient | t value | 95% CI | p value | Regression Coefficient | t value | 95% CI | p value |
| Male gender | -0.0316 | -1.97 | -0.0630 to -0.0001 | 0.049 | -0.114 | -9.19 | -0.138 to -0.089 | <0.01 |
| Age (years) | -0.00355 | -6.06 | -0.00470 to -0.00240 | <0.01 | 0.00246 | 5.01 | 0.00150 to 0.00343 | <0.01 |
| Ln Dialysis duration (months) | 0.190 | 25.6 | 0.176 to 0.205 | <0.01 | 0.0483 | 6.72 | 0.0342 to 0.0624 | <0.01 |
| Diabetes | -0.0206 | -1.19 | -0.0545 to 0.0134 | 0.23 | 0.0598 | 4.55 | 0.0340 to 0.0857 | <0.01 |
| BMI (kg/m$^2$) | -0.00373 | -1.62 | -0.00827 to 0.00080 | 0.11 | -0.0096 | -5.33 | -0.0131 to -0.0060 | <0.01 |
| History of AMI | -0.0170 | -0.52 | -0.0808 to 0.0468 | 0.60 | 0.0329 | 1.35 | -0.0148 to 0.0806 | 0.18 |
| History of cerebral hemorrhage | 0.0833 | 1.81 | -0.0068 to 0.1735 | 0.07 | 0.0314 | 0.91 | -0.0362 to 0.0989 | 0.36 |
| History of cerebral infarction | 0.0297 | 1.13 | -0.0220 to 0.0813 | 0.26 | 0.0023 | 0.11 | -0.0381 to 0.0427 | 0.91 |
| BUN (mg/dL) | 0.00160 | 3.02 | 0.00056 to 0.00265 | <0.01 | -0.00224 | -5.41 | -0.00305 to -0.00143 | <0.01 |
| Cr (mg/dL) | 0.0717 | 35.8 | 0.0678 to 0.0756 | <0.01 | 0.0728 | 30.7 | 0.0682 to 0.0775 | <0.01 |
| Alb (g/L) | -0.0754 | -4.92 | -0.1055 to -0.0454 | <0.01 | -0.0931 | -6.75 | -0.1203 to -0.0660 | <0.01 |
| Ln CRP (mg/dL) | 0.0214 | 5.13 | 0.0133 to 0.0296 | <0.01 | 0.0110 | 3.08 | 0.0040 to 0.0181 | <0.01 |
| Hb (g/dL) | -0.0617 | -11.8 | -0.0719 to -0.0514 | <0.01 | -0.0209 | -5.11 | -0.0289 to -0.0129 | <0.01 |
| D/P Cr | 0.0970 | 1.18 | -0.0638 to 0.2579 | 0.24 | -0.013 | -0.23 | -0.128 to 0.101 | 0.817 |
| Use of icodextrin | 0.251 | 12.4 | 0.211 to 0.290 | <0.01 | 0.0600 | 3.91 | 0.0298 to 0.0902 | <0.01 |
| Ln UV (mL/day) | -0.0878 | -26.9 | -0.0942 to -0.0814 | <0.01 | -0.0433 | -12.8 | -0.0500 to -0.0366 | <0.01 |
| History of PD peritonitis | 0.0475 | 2.02 | 0.0014 to 0.0936 | 0.04 | 0.0142 | 0.85 | -0.0189 to 0.0473 | 0.40 |

Abbreviations: B2M, β2 microglobulin; CI, confidence interval; BMI, body mass index; AMI, acute myocardial infarction; BUN, blood urea nitrogen; Cr, creatinine; Alb, albumin; CRP, C reactive protein; Hb, hemoglobin; D/P Cr, dialysate-to-plasma ratio of creatinine; UV, urinary volume; PD, peritoneal dialysis.

Table 2 shows the results of multiple regression analysis. After multiple imputation for missing variables, sex, age, dialysis duration, diabetes, BMI, BUN, Cr, Alb, CRP, Hb, use of icodextrin and UV were all found to be independent factors affecting serum B2M. Among them, Cr and UV were strongly associated with serum B2M (t = 30.7, p < 0.01 and t = -12.8, p < 0.01, respectively).

During a median follow-up of 87 months, 2,054 patients transferred to other dialysis modalities such as combined therapy with PD and HD (n = 697) or HD directly (n = 1,357). A total of 1,235 (41.0%) died, including 437 patients (14.5%) from CV causes. Among them, 612 patients died after transfer to other dialysis modalities (n = 163 and n = 449 for patients transferred to combined therapy with PD and HD and those transferred to HD directly, respectively). Kaplan-Meier analysis showed that neither all-cause nor CV mortality differed significantly between four groups according to serum B2M (P = 0.35, P = 0.62, respectively; Fig 2A and 2B). Serum B2M was not associated with all-cause mortality in univariate analysis, whereas higher serum B2M was independently associated with higher all-cause mortality in the adjusted model (HR 1.07; 95%CI 0.93 to 1.22 and HR 1.64; 95%CI 1.33 to 2.03, respectively) in multivariate Cox regression analyses after multiple imputation for missing variables (Table 3). In addition, male sex, higher age, longer dialysis duration, diabetes, higher BUN, lower Cr, lower Alb, higher CRP and a greater number of comorbidities, including acute myocardial infarction (AMI), cerebral bleeding and cerebral infarction were associated with higher all-cause mortality. S2 Table shows the results of the Cox regression analysis after multiple imputation for missing variables treating serum B2M as a categorical variable. An unadjusted model revealed that all-cause mortality is constant regardless of serum B2M quantile. An adjusted model reveled that higher serum B2M (24.7 to <33.3 mg/L and ≥33.3 mg/L) were independently associated with higher all-cause mortality (HR 1.33; 95%CI 1.12 to 1.58 and HR

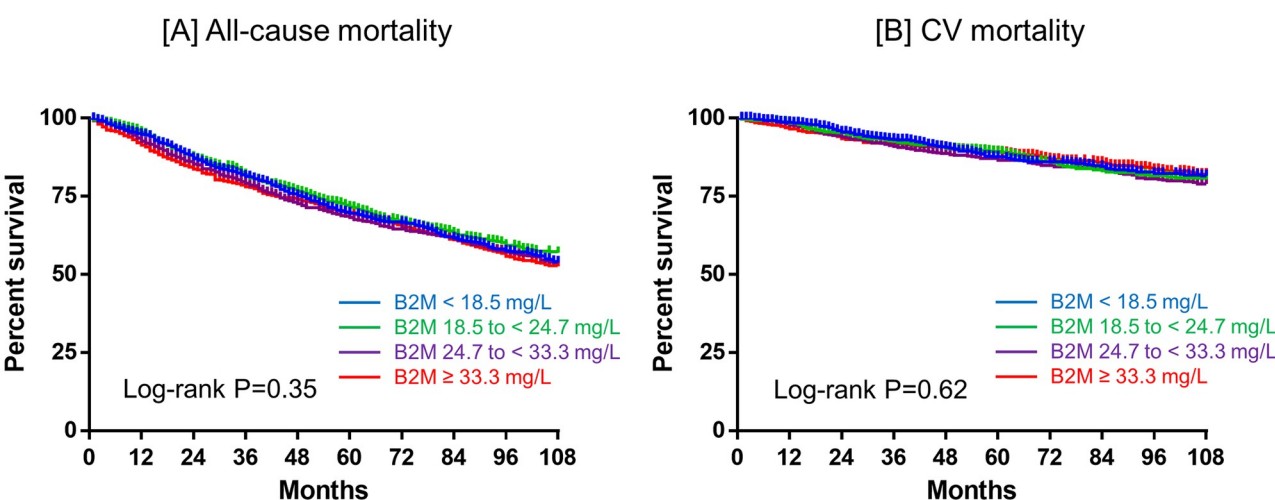

**Fig 2. Kaplan-Meier curve of all-cause (A) and CV mortality (B).** These parameters were compared between four groups according to serum B2M. Abbreviations: CV, cardiovascular; B2M, β2 microglobulin.

1.71; 95%CI 1.40 to 2.08, respectively), whereas lower serum B2M (<18.5 mg/L) did not affect all-cause mortality (HR 0.98; 95%CI 0.83 to 1.17). Fig 3 shows the results of restricted cubic spline curve. An unadjusted analysis revealed that all-cause mortality is constant regardless of serum B2M, whereas an adjusted analysis revealed a linear correlation between serum B2M

**Table 3. HR and 95% CI of all-cause mortality and SHR and 95% CI of CV mortality.**

| | HR and 95% CI of all-cause mortality | | SHR and 95% CI of CV mortality | |
| --- | --- | --- | --- | --- |
| | Unadjusted | Adjusted | Unadjusted | Adjusted |
| Male gender | 1.17 (1.04 to 1.31) | 1.46 (1.28 to 1.66) | 1.21 (0.99 to 1.47) | 1.23 (0.99 to 1.53) |
| Age [yr] | 1.08 (1.07 to 1.08) | 1.06 (1.06 to 1.07) | 1.04 (1.04 to 1.05) | 1.04 (1.03 to 1.05) |
| Ln Dialysis duration [months] | 1.03 (0.97 to 1.10) | 1.18 (1.09 to 1.27) | 0.96 (0.87 to 1.05) | 1.09 (0.96 to 1.24) |
| Diabetes | 1.66 (1.48 to 1.87) | 1.49 (1.32 to 1.69) | 1.99 (1.65 to 2.41) | 1.78 (1.45 to 2.19) |
| BMI [kg/m$^2$] | 0.98 (0.96 to 0.999) | 1.00 (0.98 to 1.02) | 1.01 (0.98 to 1.04) | 1.01 (0.98 to 1.04) |
| BUN [mg/dL] | 0.991 (0.987 to 0.994) | 1.006 (1.001 to 1.010) | 0.993 (0.987 to 0.999) | 1.000 (0.993 to 1.006) |
| Cr [mg/dL] | 0.88 (0.86 to 0.89) | 0.88 (0.85 to 0.91) | 0.92 (0.89 to 0.95) | 0.96 (0.91 to 1.01) |
| Ln B2M [mg/L] | 1.07 (0.93 to 1.22) | 1.64 (1.33 to 2.03) | 1.07 (0.86 to 1.33) | 1.48 (1.02 to 2.16) |
| Alb [g/dL] | 0.35 (0.31 to 0.39) | 0.62 (0.54 to 0.70) | 0.53 (0.44 to 0.63) | 0.81 (0.65 to 1.02) |
| Ln CRP [mg/dL] | 1.24 (1.20 to 1.28) | 1.11 (1.07 to 1.14) | 1.17 (1.11 to 1.24) | 1.08 (1.02 to 1.14) |
| Hb [g/dL] | 0.93 (0.90 to 0.97) | 0.97 (0.93 to 1.02) | 0.93 (0.87 to 0.99) | 0.96 (0.89 to 1.03) |
| History of AMI | 2.51 (2.08 to 3.03) | 1.31 (1.08 to 1.60) | 2.13 (1.57 to 2.89) | 1.34 (0.96 to 1.89) |
| History of cerebral bleeding | 1.82 (1.38 to 2.40) | 1.34 (1.003 to 1.79) | 1.68 (1.06 to 2.65) | 1.17 (0.71 to 1.95) |
| History of cerebral infarction | 2.53 (2.17 to 2.96) | 1.26 (1.07 to 1.48) | 1.70 (1.29 to 2.24) | 1.03 (0.77 to 1.38) |
| D/P Cr ratio | 2.53 (1.35 to 4.75) | 0.88 (0.46 to 1.68) | 1.63 (0.53 to 4.98) | 0.70 (0.27 to 1.84) |
| Use of icodextrin | 1.13 (0.97 to 1.32) | 1.04 (0.89 to 1.22) | 1.11 (0.86 to 1.43) | 1.03 (0.77 to 1.36) |
| Ln UV [mL/day] | 0.98 (0.96 to 1.01) | 0.98 (0.95 to 1.02) | 1.05 (0.99 to 1.11) | 1.06 (1.001 to 1.13) |
| History of PD peritonitis | 1.29 (1.09 to 1.52) | 0.93 (0.79 to 1.10) | 1.13 (0.85 to 1.51) | 0.94 (0.70 to 1.26) |

Abbreviations: HR, hazard ratio; CI, confidence interval; SHR, sub-hazard ratio; CV, cardiovascular; BMI, body mass index; BUN, blood urea nitrogen; Cr, creatinine; B2M, β2 microglobulin; Alb, albumin; CRP, C-reactive protein; Hb, hemoglobin; AMI, acute myocardial infarction; D/P Cr, dialysate-to-plasma ratio of creatinine; UV, urinary volume; PD, peritoneal dialysis.

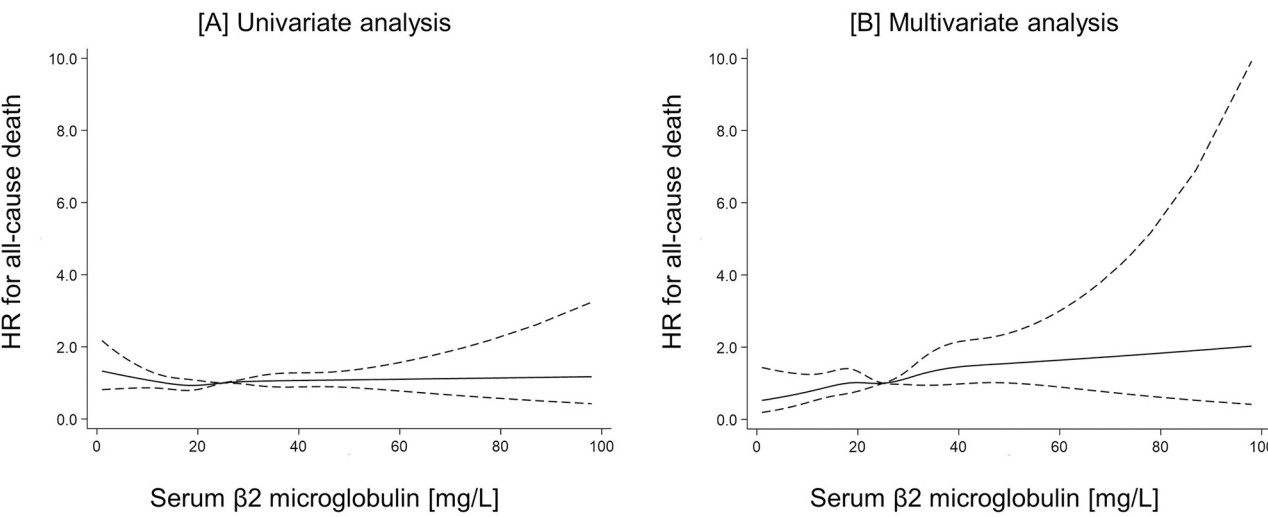

**Fig 3. HRs for all-cause mortality derived from univariate [A] and multivariate analysis [B].** Lines represent HRs and 95% CIs based on restricted cubic splines for serum B2M levels with five knots at the 5th, 27.5th, 50th, 72.5th, and 95th percentiles. Abbreviations: HR, hazard ratio.

and all-cause mortality. Serum B2M was not associated with CV mortality in univariate analysis, whereas higher serum B2M was independently associated with higher CV mortality in the adjusted model (SHR 1.07; 95%CI, 0.86 to 1.33 and SHR 1.48; 95%CI 1.02 to 2.16, respectively) in a multivariate competing-risks regression analysis after multiple imputation for missing variables (Table 3). Additionally, higher age, diabetes, higher CRP and higher UV were associated with higher CV mortality. S2 Table also shows the results of the competing-risks regression analysis after multiple imputation for missing variables treating serum B2M as a categorical variable. Both unadjusted and adjusted model revealed that CV mortality is constant regardless of serum B2M quantile.

Fig 4 shows results of subgroup analysis focused on associations between serum B2M and all-cause mortality. UV and a history of AMI significantly affected the impact of serum B2M on all-cause mortality (HR 1.63; 95%CI 1.08 to 2.48 and HR 0.84; 95%CI 0.33 to 2.17 for patients with UV ≥ 100 mL/day and those with UV < 100 mL/day, respectively, interaction P = 0.03 and HR 1.70; 95%CI 0.57 to 5.09 and HR 1.78; 95%CI 1.17 to 2.71, for patients with a history of AMI and those without a history of AMI, respectively, interaction P = 0.02). However, it should be taken into account that the number of patients with UV < 100 mL/day (n = 362) and those with a history of AMI (n = 182) were small.

We conducted subgroup Cox regression analysis for all-cause mortality among patients continued PD, those transferred to combined therapy with PD and HD and those directly transferred to HD, separately (Table 4). In univariate Cox regression analyses, higher serum B2M was associated with higher all-cause mortality only among patients continued PD (HR 1.33; 95%CI 1.12 to 1.58, HR 1.26; 95%CI 0.83 to 1.89 and HR 1.18; 95%CI 0.94 to 1.48 for patients continued PD, those transferred to combined therapy and those transferred to HD directly, respectively). In multivariate Cox regression analyses after multiple imputation for missing variables, higher serum B2M was independently associated with higher all-cause mortality in all groups (HR 1.48; 95%CI 1.10 to 2.00, HR 2.26; 95%CI 1.17 to 4.33 and HR 2.04; 95%CI 1.42 to 2.92 for patients continued PD, those transferred to combined therapy and those transferred to HD directly, respectively). Dialysis modalities did not affect the association between serum B2M and mortality (interaction P = 0.18, Fig 4). Kaplan-Meier analysis

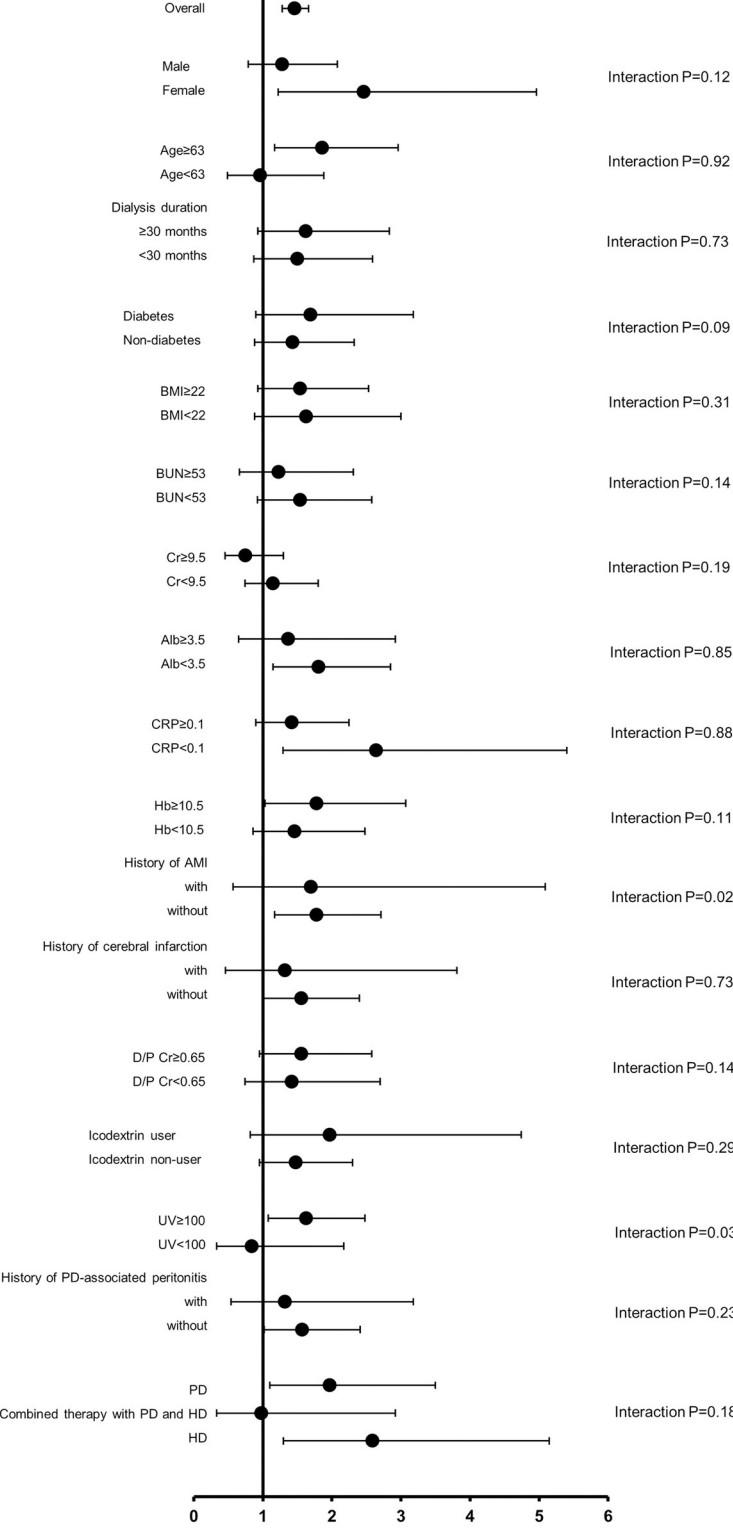

**Fig 4. Subgroup analysis for the association between serum B2M and all-cause mortality.** Dots and bars correspond to adjusted HRs and 95% CIs for all-cause death. Abbreviations: B2M, β2 microglobulin; HR, hazard ratio; CI, confidence interval; BMI, body mass index; BUN, blood urea nitrogen; Cr, creatinine; Alb, albumin; CRP, C-reactive protein; Hb, hemoglobin; AMI, acute myocardial infarction; D/P Cr, dialysate-to-plasma ratio of creatinine; UV, urinary volume; PD, peritoneal dialysis; HD, hemodialysis.

**Table 4. HR and 95% CI of all-cause for each dialysis modality.**

| | Patients continued PD | | Patients transferred to combined therapy | | Patients directly transferred to HD | |
|---|---|---|---|---|---|---|
| | Unadjusted | Adjusted | Unadjusted | Adjusted | Unadjusted | Adjusted |
| Male gender | 1.35 (1.15 to 1.58) | 1.31 (1.10 to 1.57) | 1.76 (1.23 to 2.53) | 1.96 (1.32 to 2.93) | 1.26 (1.04 to 1.53) | 1.60 (1.28 to 1.99) |
| Age [yr] | 1.04 (1.03 to 1.05) | 1.03 (1.03 to 1.04) | 1.08 (1.06 to 1.10) | 1.08 (1.06 to 1.10) | 1.08 (1.07 to 1.09) | 1.07 (1.06 to 1.08) |
| Ln Dialysis duration [months] | 1.04 (0.96 to 1.12) | 1.12 (1.02 to 1.24) | 1.08 (0.91 to 1.29) | 1.16 (0.93 to 1.45) | 1.04 (0.94 to 1.15) | 1.25 (1.09 to 1.43) |
| Diabetes | 1.59 (1.35 to 1.88) | 1.41 (1.18 to 1.70) | 1.84 (1.34 to 2.54) | 1.53 (1.06 to 2.22) | 1.76 (1.46 to 2.13) | 1.69 (1.37 to 2.09) |
| BMI [kg/m$^2$] | 1.02 (0.997 to 1.05) | 1.02 (0.99 to 1.05) | 1.04 (0.996 to 1.09) | 1.04 (0.98 to 1.10) | 0.98 (0.95 to 1.01) | 0.98 (0.95 to 1.01) |
| BUN [mg/dL] | 0.999 (0.993 to 1.004) | 1.005 (0.999 to 1.011) | 1.006 (0.995 to 1.017) | 1.013 (0.999 to 1.025) | 0.996 (0.989 to 1.002) | 1.012 (1.004 to 1.019) |
| Cr [mg/dL] | 0.96 (0.93 to 0.98) | 0.96 (0.91 to 1.01) | 0.94 (0.90 to 0.99) | 0.90 (0.83 to 0.98) | 0.89 (0.86 to 0.92) | 0.86 (0.81 to 0.90) |
| Ln B2M [mg/L] | 1.33 (1.12 to 1.58) | 1.48 (1.10 to 2.00) | 1.26 (0.83 to 1.89) | 2.26 (1.17 to 4.33) | 1.18 (0.94 to 1.48) | 2.04 (1.42 to 2.92) |
| Alb [g/dL] | 0.39 (0.34 to 0.45) | 0.57 (0.48 to 0.69) | 0.65 (0.45 to 0.93) | 0.89 (0.60 to 1.30) | 0.39 (0.32 to 0.47) | 0.64 (0.50 to 0.81) |
| Ln CRP [mg/dL] | 1.20 (1.15 to 1.25) | 1.06 (1.01 to 1.11) | 1.20 (1.11 to 1.31) | 1.14 (1.04 to 1.24) | 1.22 (1.16 to 1.28) | 1.12 (1.06 to 1.18) |
| Hb [g/dL] | 0.89 (0.84 to 0.93) | 0.95 (0.90 to 1.01) | 0.95 (0.85 to 1.06) | 1.02 (0.90 to 1.16) | 0.92 (0.86 to 0.98) | 0.94 (0.88 to 1.02) |
| History of AMI | 2.01 (1.55 to 2.59) | 1.38 (1.06 to 1.80) | 2.36 (1.34 to 4.17) | 1.30 (0.69 to 2.43) | 2.67 (1.92 to 3.72) | 1.20 (0.84 to 1.71) |
| History of cerebral bleeding | 1.88 (1.31 to 2.70) | 1.22 (0.82 to 1.81) | 2.45 (1.15 to 5.24) | 2.09 (0.92 to 4.75) | 1.29 (0.76 to 2.20) | 1.11 (0.64 to 1.93) |
| History of cerebral infarction | 2.15 (1.75 to 2.64) | 1.34 (1.08 to 1.67) | 2.36 (1.42 to 3.91) | 1.36 (0.77 to 2.39) | 2.06 (1.55 to 2.73) | 1.04 (0.77 to 1.40) |
| D/P Cr ratio | 3.04 (1.29 to 7.15) | 1.12 (0.48 to 2.64) | 0.45 (0.10 to 2.01) | 0.55 (0.11 to 2.87) | 4.69 (1.61 to 13.69) | 0.86 (0.31 to 2.42) |
| Use of icodextrin | 1.33 (1.07 to 1.64) | 0.97 (0.77 to 1.23) | 1.14 (0.76 to 1.72) | 1.04 (0.67 to 1.61) | 1.08 (0.84 to 1.39) | 1.07 (0.81 to 1.41) |
| Ln UV [mL/day] | 0.97 (0.94 to 1.003) | 0.99 (0.95 to 1.04) | 1.00 (0.92 to 1.07) | 1.02 (0.92 to 1.13) | 1.02 (0.97 to 1.07) | 1.01 (0.95 to 1.08) |
| History of PD peritonitis | 1.30 (1.03 to 1.64) | 1.08 (0.87 to 1.36) | 1.22 (0.74 to 2.00) | 0.93 (0.56 to 1.54) | 1.32 (1.01 to 1.71) | 0.90 (0.69 to 1.17) |

Abbreviations: HR, hazard ratio; CI, confidence interval; PD, peritoneal dialysis; HD, hemodialysis; BMI, body mass index; BUN, blood urea nitrogen; Cr, creatinine; B2M, β2 microglobulin; Alb, albumin; CRP, C-reactive protein; Hb, hemoglobin; AMI, acute myocardial infarction; D/P Cr, dialysate-to-plasma ratio of creatinine; UV, urinary volume.

involving only patients continued PD revealed that all-cause mortality in the higher B2M group was significantly higher compared to that in the lower B2M groups especially during 36 months (log-rank P = 0.02; Fig 5).

## Discussion

This observational study using a large-scale registry of 3,011 Japanese PD patients over a 9-year follow-up revealed that higher serum B2M was independently associated with all-cause mortality. The associations between higher serum B2M and higher all-cause mortality were already certified in both non-dialyzed CKD [9–11] and HD patients [12]. However, the findings contrast with those of previous studies conducted in PD patients. Koh et al. [20] reported that higher serum B2M was associated with higher all-cause mortality among 771 Korean prevalent PD patients. However, the effect of higher serum B2M disappeared after adjusting for residual renal clearance, calculated as the mean of creatinine and urea clearances. Chang et al. [21] also reported that an association between higher serum B2M and higher mortality disappeared after adjusting for residual renal clearance applying the same method among 725 Korean incident PD patients. Conversely, the increased risk of all-cause death in patients with lower serum B2M became more evident even after adjusting for several confounding factors including RKF. RKF is well known as a strong predictor of mortality among PD patients [27, 28], and B2M can accurately estimate RKF in dialyzed patients [7, 8, 16, 17]. Indeed, UV was the strongest independent factor affecting serum B2M in our multiple regression analysis. In addition, a positive association between higher serum B2M and higher mortality was more

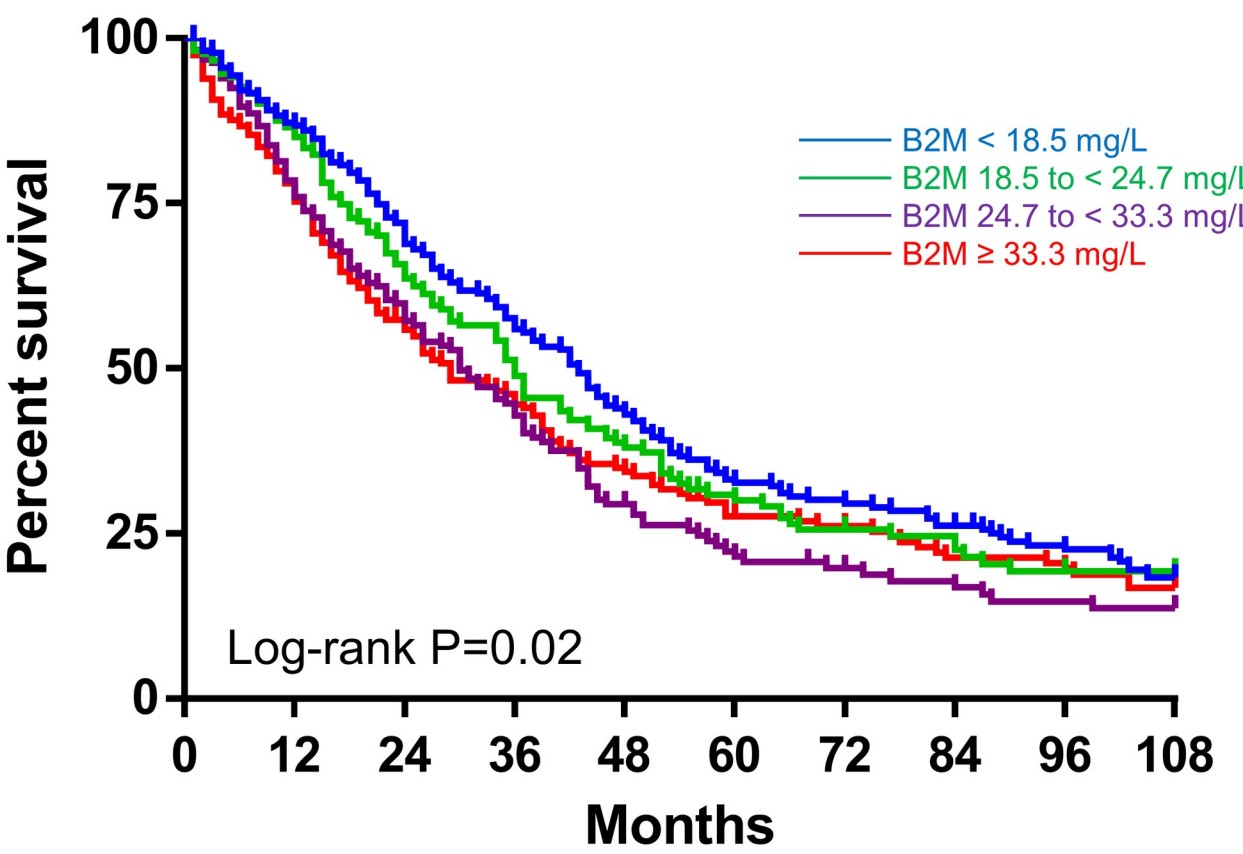

**Fig 5. Kaplan-Meier curve of all-cause mortality among patients continued PD.** This parameter was compared between four groups according to serum B2M. Abbreviations: B2M, β2 microglobulin.

prominent among patients with preserved RKF and there was a significant interaction, as well as the results of previous two studies [20, 21]. Based on the above, RKF is thought to be most important factor in the effect of serum B2M on mortality.

Meanwhile, serum B2M was not associated with all-cause or CV mortality in our univariate analyses. We speculated that higher serum B2M is associated with higher mortality, because it reflects not only the accumulation of middle molecular weight uremic toxins, but also inflammation or declines in RKF. Although the mechanisms responsible for this observation of univariate analysis remain unclear, malnutrition and immunological disturbance might be involved. B2M is synthesized and shed mainly by lymphocytes [29, 30], and its count is one of the most commonly used nutritional parameters [31]. However, no definitive proof of alterations in the generation or metabolism of B2M has been seen from nutritional status, especially in dialyzed patients. Given the possibility that the association between serum B2M and mortality is nonlinear, we conducted multivariate analysis treating serum B2M as a categorical variable and analysis using restricted cubic spline curve and found that there was no U-shaped association. At least, this finding can lead to the consequence that serum B2M *per se* had no predictive value for patient outcome in PD patients. We conducted subgroup analysis for all-cause mortality among patients continued PD, those transferred to combined therapy with PD and HD and those directly transferred to HD, separately, and found that higher serum B2M was associated with higher all-cause mortality in univariate analysis only among patients continued PD. Although these data could suggest that serum B2M reflects patient survival before

transfer to other dialysis modality, further study is needed to clarify the effects of serum B2M on mortality in PD patients.

In multiple regression analysis, Cr and dialysis duration were strongly associated with serum B2M. Additionally, both were independent prognostic factor for all-cause death in multivariate Cox regression analysis. Cr was reported to be positively correlated with serum B2M not only in non-dialyzed patients [32, 33] but also in dialyzed patients [16, 20]. Recent systematic review including four studies of dialyzed patients, two focused on PD and two focused on HD, revealed that higher circulating B2M was associated significantly with higher risk of all-cause mortality [34], whereas several large cohort studies of dialyzed patients revealed that Cr was inversely correlated with mortality risk [35, 36]. This discrepancy could be explained by reverse epidemiology [37]. Since Cr reflects muscle mass or meat ingestion and/or the degree of dialysis efficiency in dialyzed patients, harmful effect of higher Cr in the general population is completely reversed in patients on dialysis. On the other hand, an independent association between serum B2M and dialysis duration has been already demonstrated both in PD [20] and HD patients [12]. Additionally, these studies revealed that longer dialysis duration was independently associated with higher mortality. Although the detailed mechanisms are unclear, it is speculated that not only RKF declines but also inflammation, malnutrition, left ventricular hypertrophy or immune dysfunction become prominent in patients with longer dialysis duration and therefore longer dialysis duration could be independently associated with poor patient outcome even after adjustment for known risk factors.

Strengths of our study include the examination of a large cohort of PD patients, allowing for more definite results with extensive adjustments and subgroup analyses. However, several limitations to the study should be discussed. First, the observational design allowed only limited conclusions. In particular, we cannot prove any cause-and-effect relationships. Second, serum B2M and other laboratory data were measured only at baseline, so we could not determine the effects of changes from baseline during follow-up using time-dependent analyses. Third, we used UV as a surrogate marker of RKF. Since UV has large variation from dietary intake or non-renal output, renal clearance of urea and creatinine could better reflect RKF. Additionally, one-third of enrolled patients did not have UV data. However, we could not use Kt/V as confounding factor because approximately two-thirds of patients did not have this data.

## Conclusion

Using a large-scale registry of 3,011 Japanese PD patients, higher serum B2M was independently associated with higher all-cause mortality especially among patients with preserved UV. These data suggest that B2M contributes significantly to worse patient outcome and RKF significantly affects this relationship. On the contrary, serum B2M *per se* had no predictive value for patient outcome in this population. Further investigations are needed to validate the clinical utility of serum B2M among PD patients.

## Supporting information

**S1 Table. Comparisons of baseline characteristics between patients with and without serum B2M data.**
(DOCX)

**S2 Table. HR and 95% CI of all-cause mortality and SHR and 95% CI of CV mortality, with analysis using serum B2M as a categorical variable.**
(DOCX)

## Acknowledgments

The authors thank the participants in the dialysis registry of the JSDT, the Committee of the Renal Data Registry, and all personnel at the institutions that participated in this survey. The data used in this study were provided by JSDT. The interpretation and reporting of these data are the responsibility of the authors, which is in no way seen as an official policy or interpretation of the JSDT.

## Author Contributions

**Conceptualization:** Yukio Maruyama, Masaaki Nakayama, Masanori Abe, Takashi Yokoo, Jun Minakuchi, Kosaku Nitta.

**Data curation:** Yukio Maruyama.

**Formal analysis:** Yukio Maruyama.

**Investigation:** Yukio Maruyama.

**Methodology:** Yukio Maruyama, Masaaki Nakayama, Masanori Abe.

**Supervision:** Masaaki Nakayama, Masanori Abe, Takashi Yokoo, Jun Minakuchi, Kosaku Nitta.

**Writing – original draft:** Yukio Maruyama.

**Writing – review & editing:** Masaaki Nakayama, Masanori Abe, Takashi Yokoo, Jun Minakuchi, Kosaku Nitta.

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
