## [Decision Letter · Decision Letter 0]

26 Jan 2022

PONE-D-22-00429Association between serum β2-microglobulin and mortality in Japanese peritoneal dialysis patients: a cohort studyPLOS ONE

Dear Dr. Maruyama,

Thank you for submitting your manuscript to PLOS ONE. After careful consideration, we feel that it has merit but does not fully meet PLOS ONE’s publication criteria as it currently stands. Therefore, we invite you to submit a revised version of the manuscript that addresses the points raised during the review process. Please have a look at additional editor comments to the authors.

The reviewers proposed several important queries. Please respond to all of the concerns.

We look forward to receiving your revised manuscript.

Kind regards,

Kojiro Nagai

Academic Editor

PLOS ONE

Journal Requirements:

"I have read the journal's policy and the authors of this manuscript have the following competing interests: Y.M. received honoraria from Baxter International, Inc. and Terumo Corporation. No other authors have any conflicts of interest to declare."

We note that you received funding from a commercial source: Baxter International, Inc. and Terumo Corporation

Within this Competing Interests Statement, please confirm that this does not alter your adherence to all PLOS ONE policies on sharing data and materials by including the following statement: ""This does not alter our adherence to PLOS ONE policies on sharing data and materials.” (as detailed online in our guide for authors http://journals.plos.org/plosone/s/competing-interests).  If there are restrictions on sharing of data and/or materials, please state these. Please note that we cannot proceed with consideration of your article until this information has been declared. 

Additional Editor Comments:

Is it OK to conclude that serum b2M was independently associated with CV mortality?

I do understand the efforts to analyze the relationship between b2M and mortality in a large scale cohort by using multiple methods of analysis.

However, it seems that the association between b2M and CV mortality was not constant among these methods (S2 Table).

In “limitation”, it is better to describe that one third of enrolled patients did not have UV data.

Reviewers' comments:

Reviewer's Responses to Questions

**Comments to the Author**

1. Is the manuscript technically sound, and do the data support the conclusions?

Reviewer #1: Partly

Reviewer #2: Yes

2. Has the statistical analysis been performed appropriately and rigorously? 

Reviewer #1: No

Reviewer #2: Yes

3. Have the authors made all data underlying the findings in their manuscript fully available?

Reviewer #1: Yes

Reviewer #2: Yes

4. Is the manuscript presented in an intelligible fashion and written in standard English?

Reviewer #1: Yes

Reviewer #2: Yes

5. Review Comments to the Author

Reviewer #1: The authors revealed that B2M levels contributes significantly to all-cause and CV mortality, and residual kidney function significantly affect this relationship in patients with PD. These findings were reasonable, but several revisions would be required.

1. The authors should show Kaplan-Meier curve for B2M levels.

2. Higher serum B2M levels was shown as a significant risk factor for mortality, but the impact seems to be not big, according to the results from Figure 2 and Table 3. For example, if compare 1st and 3rd quartile for B2M levels (18.5 mg/l and 33.3 mg/l, respectively), Ln(18.5) = 2.91 and Ln(33.3) = 3.50, hence change of drastic B2M levels increased event risk only 1.35 fold (according to the results from Table 3). How do the authors think about it?

3. It was a bit strange about relationship among serum creatinine levels, serum B2M levels and all-cause mortality risk. Higher serum creatinine levels was related to higher B2M levels strongly, but lower creatinine levels was a higher all-cause mortality risk although higher B2M levels was a higher all-cause mortality risk. Please describe your consideration in the discussion part.

4. The authors described that serum B2M levels was associated with all-cause mortality among patients with UV > 100 ml/day and patients without a history of AMI (Fig 3). However, I think it is difficult to conclude these results. The 95% CI range for patients with history of AMI was quite wide; I guess very small number of patients had the history. Also, wide range of 95% CI was overlap between patients with UV > 100 ml/day and with UV < 100 ml/day. Interaction P for UV and history of AMI seems to be small, but multiplicity of testing should be considered. Can the authors show number of patients and events in each group?

Reviewer #2: Maruyama et al. reported a higher mortality rate in peritoneal dialysis patients with high serum β2MG levels. Although the present manuscript is important in examining the significance of β2MG levels in peritoneal dialysis patients, more concise analysis of its interpretation is needed.

1. Kaplan-Meier curves for the four groups should be developed and analyzed for mortality and CV events.

2. Since the β2MG level is affected by the duration of dialysis, the duration of dialysis seems to define the mortality and CV events after all. The authors should discuss the association between serum β2MG and dialysis duration.

3. If the analysis includes the group of patients who changed to HD+PD combination therapy or HD alone and the group of patients who treated with PD alone, it is necessary to analyze them separately.

4. Why is there no association between β2MG level and all-cause mortality in the group with UV less than 100 ml in Figure 3?

6. PLOS authors have the option to publish the peer review history of their article (what does this mean?). If published, this will include your full peer review and any attached files.

Reviewer #1: No

Reviewer #2: No

---

## [Author Response · Author response to Decision Letter 0]

8 Mar 2022

Response to Reviewers

The comments from the reviewers proved very helpful. We have worked through each of the comments to address the issues raised, and have made detailed point-by-point responses to each of the comments provided by the reviewers. All authors have contributed to this revised manuscript and agree to this re-submission.

First of all, we revised all results, Tables and Figures because some erroneous data of underlying disease were discovered in original database. 

Editor Comments:

Is it OK to conclude that serum b2M was independently associated with CV mortality?

I do understand the efforts to analyze the relationship between b2M and mortality in a large scale cohort by using multiple methods of analysis.

However, it seems that the association between b2M and CV mortality was not constant among these methods (S2 Table).

RESPONSE: Thank you for raising this important point. As suggested, the effect of higher serum B2M on all-cause mortality was statistically significant but not intense. On the other hand, the significant association between serum B2M and CV mortality was found only in analysis treating serum B2M as a continuous variable and disappeared in analysis treating serum B2M as a categorical variable. We have added mention of this in the Abstract and Discussion. 

In “limitation”, it is better to describe that one third of enrolled patients did not have UV data.

RESPONSE: Thank you for a helpful comment. As suggested, we have added the following sentences to the Limitations section: “Additionally, approximately one-third of patients did not have UV data.” 

Reviewer #1: The authors revealed that B2M levels contributes significantly to all-cause and CV mortality, and residual kidney function significantly affect this relationship in patients with PD. These findings were reasonable, but several revisions would be required.

1. The authors should show Kaplan-Meier curve for B2M levels.

RESPONSE: Thank you for this helpful comment. As suggested, we conducted new analysis using Kaplan-Meier curve and confirmed that neither all-cause nor CV mortality differed significantly between four groups according to serum B2M (log-rank P=0.35, log-rank P=0.62, respectively; Fig 2A and 2B).

2. Higher serum B2M levels was shown as a significant risk factor for mortality, but the impact seems to be not big, according to the results from Figure 2 and Table 3. For example, if compare 1st and 3rd quartile for B2M levels (18.5 mg/l and 33.3 mg/l, respectively), Ln(18.5) = 2.91 and Ln(33.3) = 3.50, hence change of drastic B2M levels increased event risk only 1.35 fold (according to the results from Table 3). How do the authors think about it?

RESPONSE: Thank you for raising this important point. As Editor has also suggested, the effect of higher serum B2M on mortality was statistically significant but not intense. Multivariate analysis treating serum B2M as a categorical variable and analysis using restricted cubic spline curve were conducted toward the detection of nonlinear or U-shaped association, and we confirmed a linear association. The reason for a low impact of serum B2M on mortality is at present unclear. However, we believed that our study included one of the largest cohort of PD patients, allowing for more definite results with extensive adjustments. We have revised Conclusion both in Abstract and text. 

3. It was a bit strange about relationship among serum creatinine levels, serum B2M levels and all-cause mortality risk. Higher serum creatinine levels was related to higher B2M levels strongly, but lower creatinine levels was a higher all-cause mortality risk although higher B2M levels was a higher all-cause mortality risk. Please describe your consideration in the discussion part.

RESPONSE: Thank you for this helpful comment. We carefully checked the results of regression analysis and confirmed that Cr was positively associated with serum B2M. This finding is in accordance with previous reports. Cr was reported to be positively correlated with serum B2M not only in non-dialyzed patients [Wu HC et al. J Clin Lab Anal 31:e22056,2017 and Juraschek SP et al. Clin J Am Soc Nephrol 8:584-92,2013] but also in dialyzed patients [Koh ES et al. Am J Nephrol 42:91-8,2015 and Shafi T et al. Kidney Int 89:1099-110,2016]. Recent systematic review including four studies of dialyzed patients, two focused on PD and two focused on HD, revealed that higher circulating B2M was associated significantly with higher risk of all-cause mortality [Zhang J et al. Ther Apher Dial Online ahead of print], whereas several large cohort studies of dialyzed patients revealed that Cr was inversely correlated with mortality risk [Fink JC et al. Am J Kidney Dis 34:694-701,1999, Xue JL et al. Kidney Int 61:741-6,2002]. This discrepancy could be explained by reverse epidemiology [Kalantar-Zadeh K et al. Kidney Int 63:793-808,2003]. Since Cr reflects muscle mass or meat ingestion and/or the degree of dialysis efficiency in dialyzed patients, harmful effect of higher Cr in the general population is completely reversed in patients on dialysis. We have added this to the Discussion. 

4. The authors described that serum B2M levels was associated with all-cause mortality among patients with UV > 100 ml/day and patients without a history of AMI (Fig 3). However, I think it is difficult to conclude these results. The 95% CI range for patients with history of AMI was quite wide; I guess very small number of patients had the history. Also, wide range of 95% CI was overlap between patients with UV > 100 ml/day and with UV < 100 ml/day. Interaction P for UV and history of AMI seems to be small, but multiplicity of testing should be considered. Can the authors show number of patients and events in each group?

RESPONSE: Thank you for raising this important point. As suggested, assessing the significance of HR for mortality is inappropriate, because the number of patients with UV 100 mL/day (n= 362) and those with a history of AMI (n=182) were small. Since the chief aim of this analysis was to determine the effect of several confounding factors on the association between serum B2M and mortality, we totally revised the description of the results of subgroup analysis as follows: “UV and a history of AMI significantly affected the impact of serum B2M on all-cause mortality (HR 1.63; 95%CI 1.08 to 2.48 and HR 0.84; 95%CI 0.33 to 2.17 for patients with UV ≥ 100 mL/day and those with UV < 100 mL/day, respectively, interaction P = 0.03 and HR 1.70; 95%CI 0.57 to 5.09 and HR 1.78; 95%CI 1.17 to 2.71, for patients with a history of AMI and those without a history of AMI, respectively, interaction P = 0.02). However, it should be taken into account that the number of patients with UV < 100 mL/day (n= 362) and those with a history of AMI (n=182) were small.” 

Reviewer #2: Maruyama et al. reported a higher mortality rate in peritoneal dialysis patients with high serum β2MG levels. Although the present manuscript is important in examining the significance of β2MG levels in peritoneal dialysis patients, more concise analysis of its interpretation is needed.

1. Kaplan-Meier curves for the four groups should be developed and analyzed for mortality and CV events.

RESPONSE: Thank you for this helpful comment. As the other Reviewer also suggested, we conducted new analysis using Kaplan-Meier curve and confirmed that neither all-cause nor CV mortality differed significantly between four groups according to serum B2M (log-rank P=0.35, log-rank P=0.62, respectively; Fig 2A and 2B).

2. Since the β2MG level is affected by the duration of dialysis, the duration of dialysis seems to define the mortality and CV events after all. The authors should discuss the association between serum β2MG and dialysis duration.

RESPONSE: Thank you for this helpful comment. An independent association between serum B2M and dialysis duration has been already demonstrated both in PD [Koh ES et al. Am J Nephrol 42:91-8,2015] and HD patients [Cheung AK et al. J Am Soc Nephrol 17:546–55,2006]. Additionally, these studies revealed that longer dialysis duration was independently associated with higher mortality. Although the detailed mechanisms are unclear, it is speculated that not only RKF declines but also inflammation, malnutrition, left ventricular hypertrophy or peritoneal deterioration become prominent in patients with longer dialysis duration and therefore longer dialysis duration was independently associated with poor patient outcome even after adjustment for known risk factors. We have added this to the Discussion. 

3. If the analysis includes the group of patients who changed to HD+PD combination therapy or HD alone and the group of patients who treated with PD alone, it is necessary to analyze them separately.

RESPONSE: Thank you for raising this important point. We completely agree with the idea that dialysis modalities were important factors contributing to the association between serum B2M and mortality. During a median follow-up of 87 months, 2,053 patients transferred to other dialysis modalities such as combined therapy with PD and HD (n=697) or HD directly (n=1,357). A total of 1,235 (41.0%) died, including 437 patients (14.5%) from CV causes. Among them, 612 patients died after transfer to other dialysis modalities (n=163 and n=449 for patients transferred to combined therapy with PD and HD and those transferred to HD directly, respectively). As suggested, we conducted subgroup Cox regression analysis for all-cause mortality among patients continued PD, those transferred to combined therapy with PD and HD and those directly transferred to HD, separately (Table 4). In univariate Cox regression analyses, higher serum B2M was associated with higher all-cause mortality only among patients continued PD (HR 1.33; 95%CI 1.12 to 1.58, HR 1.26; 95%CI 0.83 to 1.89 and HR 1.18; 95%CI 0.94 to 1.48 for patients continued PD, those transferred to combined therapy and those transferred to HD directly, respectively). In multivariate Cox regression analyses after multiple imputation for missing variables, serum B2M was independently associated with higher all-cause mortality in all groups (HR 1.48; 95%CI 1.10 to 2.00, HR 2.26; 95%CI 1.17 to 4.33 and HR 2.04; 95%CI 1.42 to 2.92 for patients continued PD, those transferred to combined therapy and those transferred to HD directly, respectively). Dialysis modalities did not affect the association between serum B2M and mortality (interaction P=0.18, Fig 4). Kaplan-Meier analysis involving only patients continued PD revealed that all-cause mortality in the higher β2-M group was significantly higher compared to that in the lower β2-M groups especially during 36 months (log-rank P=0.02, Fig 5). Although these data could suggest that serum B2M reflects patient survival before transfer to other dialysis modality, further study is needed to clarify the effects of serum B2M on mortality in PD patients. We have added this to the Results and the Discussion. 

4. Why is there no association between β2MG level and all-cause mortality in the group with UV less than 100 ml in Figure 3?

RESPONSE: Thank you for raising this important point. As other Reviewer suggested, assessing the significance of HR for mortality in subgroup analysis was inappropriate, because the 95% CI range for patients with UV < 100 mL/day was quite wide because of small number of patients (n=362). Since the chief aim of this analysis was to determine the effect of several confounding factors on the association between serum B2M and mortality, we totally revised the description of the results of subgroup analysis as follows: “UV and a history of AMI significantly affected the impact of serum B2M on all-cause mortality (HR 1.63; 95%CI 1.08 to 2.48 and HR 0.84; 95%CI 0.33 to 2.17 for patients with UV ≥ 100 mL/day and those with UV < 100 mL/day, respectively, interaction P = 0.03 and HR 1.70; 95%CI 0.57 to 5.09 and HR 1.78; 95%CI 1.17 to 2.71, for patients with a history of AMI and those without a history of AMI, respectively, interaction P = 0.02). However, it should be taken into account that the number of patients with UV < 100 mL/day (n= 362) and those with a history of AMI (n=182) were small.”

---

## [Decision Letter · Decision Letter 1]

30 Mar 2022

Association between serum β2-microglobulin and mortality in Japanese peritoneal dialysis patients: a cohort study

PONE-D-22-00429R1

Dear Dr. Maruyama,

We’re pleased to inform you that your manuscript has been judged scientifically suitable for publication and will be formally accepted for publication once it meets all outstanding technical requirements.

Kind regards,

Kojiro Nagai

Academic Editor

PLOS ONE

Additional Editor Comments (optional):　All comments have been addressed.

Reviewers' comments:

Reviewer's Responses to Questions

**Comments to the Author**

1. If the authors have adequately addressed your comments raised in a previous round of review and you feel that this manuscript is now acceptable for publication, you may indicate that here to bypass the “Comments to the Author” section, enter your conflict of interest statement in the “Confidential to Editor” section, and submit your "Accept" recommendation.

Reviewer #1: All comments have been addressed

Reviewer #2: All comments have been addressed

2. Is the manuscript technically sound, and do the data support the conclusions?

Reviewer #1: (No Response)

Reviewer #2: Yes

3. Has the statistical analysis been performed appropriately and rigorously? 

Reviewer #1: (No Response)

Reviewer #2: Yes

4. Have the authors made all data underlying the findings in their manuscript fully available?

Reviewer #1: (No Response)

Reviewer #2: Yes

5. Is the manuscript presented in an intelligible fashion and written in standard English?

Reviewer #1: (No Response)

Reviewer #2: Yes

6. Review Comments to the Author

Reviewer #1: All comments have been addressed. I have no further comments. I think the manuscript would be acceptable.

Reviewer #2: The authors revised nicely. This research will cotribute to understand the status of PD patients.

7. PLOS authors have the option to publish the peer review history of their article (what does this mean?). If published, this will include your full peer review and any attached files.

Reviewer #1: No

Reviewer #2: No

---

## [Editor Report · Acceptance letter]

6 Apr 2022

PONE-D-22-00429R1 

Association between serum β2-microglobulin and mortality in Japanese peritoneal dialysis patients: a cohort study 

Dear Dr. Maruyama:

I'm pleased to inform you that your manuscript has been deemed suitable for publication in PLOS ONE. Congratulations! Your manuscript is now with our production department. 

Kind regards, 

on behalf of

Dr. Kojiro Nagai 

Academic Editor

PLOS ONE